# From Sedentary and Physical Inactive Behaviours to an Ultra Cycling Race: A Mixed-Method Case Report

**DOI:** 10.3390/ijerph17020502

**Published:** 2020-01-13

**Authors:** Kenny Guex, Sophie Wicht, Cyril Besson, Francis Degache, Boris Gojanovic, Gerald Gremion

**Affiliations:** 1School of Health Sciences (HESAV), HES-SO University of Applied Sciences and Arts Western Switzerland, 1011 Lausanne, Switzerland; kenny.guex@hesav.ch (K.G.); sophie.wicht@hesav.ch (S.W.); 2Department of Sports Medicine, Lausanne University Hospital, 1011 Lausanne, Switzerland; cyril.besson@chuv.ch (C.B.); gerald.gremion@chuv.ch (G.G.); 3Therapeutic and Performance Sports Institute, MotionLab, 1052 Le Mont-sur-Lausanne, Switzerland; 4Swiss Olympic Medical Center, Hôpital de La Tour, 1217 Meyrin, Switzerland; boris.gojanovic@latour.ch; 5Centre SportAdo, Département Femme-Mère-Enfant (DFME), CHUV et Université de Lausanne, 1011 Lausanne, Switzerland

**Keywords:** physical activity, sedentary, health, challenge, endurance training, strength training

## Abstract

In faculties of health sciences, almost 30% of nursing students exercise less than once a week. This mixed-method case report presents the 38-month evolution of the physiological and psychological health parameters of a sedentary and physically inactive nursing student. During this period, she first took part in a one-semester institutional physical activity (PA) program that was offered by her university before being selected for participation in the Race Across America (RAAM) with a university relay team. In the four months leading up to the RAAM, she followed a cycling training program. After the RAAM, she was followed-up for the next 28 months. The results showed that each phase of the study had an important impact on the subject and showed that sedentary and physical inactive behaviours are reversible. Institutional PA programs, including training education in addition to concurrent strength and endurance training, can lead to physiological and psychological health improvements. For some individuals, participating in an athletic challenge can improve motivation and long-term adherence to PA participation. An individualised approach should be considered in future interventions that aim to promote PA participation. In the specific context of a university of health sciences, this kind of initiative could positively influence the general population’s health by empowering students to become role models towards PA promotion.

## 1. Introduction

A majority of our time awake is spent being sedentary [1]. Sedentary behaviour is defined as any non-sleep behaviour done in a sitting, reclining or lying posture that requires an energy expenditure ≤1.5 metabolic equivalents (MET) [2,3]. It is well-established that sedentary behaviour increases the risk of multiple adverse health conditions and early mortality [4,5]. Physical inactivity differs from sedentary behaviour, in that it is defined as an insufficient physical activity (PA) level to meet recommendations [2,3]: at least 150 min/week of moderate-intensity aerobic PA, or at least 75 min/week of vigorous-intensity aerobic PA, or an equivalent combination of moderate- and vigorous-intensity PA [6]. Failing to meet these recommendations is responsible for more than five million deaths worldwide each year [4], whereas meeting them decreases the mortality risk associated with high sitting time [7].

Sedentary behaviour is an important issue for university students, as they spend most of their time sitting for studying [8]. However, sedentary and physically active behaviours appear largely uncorrelated, since students could also be highly active [8,9]. In faculties of health sciences, almost 30% of nursing students exercise less than once a week [10], leading to a lower maximal oxygen uptake (VO_2_max) than a general academic population (35.7 vs. 41.3 mL/kg/min) [11]. Aside from being beneficial for health, PA might condition one’s own health behaviour and one’s role as a health-promoting actor. Indeed, nurses who are more physically active and who receive formal training in this field have been shown to better promote PA in their clinical practice [12].

In order to improve global health outcomes for its student population, a university of health sciences in Switzerland developed a one-semester PA program for physically inactive students. As part of its novel PA promotion strategy, the university registered to participate to the Race Across America (RAAM) in a team of eight cyclists nine months after the beginning of this program. The team was composed of professors and students, including a sedentary student who completed the institutional PA program [13]. The RAAM is among the longest and most difficult ultra-endurance cycling races in the world [14,15]. It consists of cycling in one uninterrupted stage ~4900 km from the west to the east coast.

The aim of the present mixed-method case report was to present the physiological and psychological health parameters of a sedentary and physically inactive student who participated in the RAAM with the university team. In order to better understand and interpret the participant’s experience, we used a qualitative approach at the end of a follow-up period of 28 months after the RAAM.

## 2. Materials and Methods

We reported on a 30-year-old (168 cm, 67.9 kg) female nursing sciences student who engaged in no exercise all at the time of inclusion and had never ridden a bike. The study was conducted over 38 months and was composed of four periods (Figure 1). At the beginning of the second year of her bachelor’s degree, she registered to participate to the one-semester institutional PA program that was offered by the university. Thanks to her adherence, progression, and high degree of motivation, she was selected to participate in the university team’s for the RAAM. During the next four months, she followed the same specific cycling training program as the seven other members of the team and then took part to the RAAM. Finally, she was followed-up for the next 28 months. We collected physiological and psychological health parameters prior to the beginning (T1) and at the end (T2) of the institutional PA program, directly after the specific cycling training program (T3) and the RAAM (T4). Follow-up tests were performed at two (T5) and 28 months (T6) after the RAAM in order to monitor the participant’s progression one year after the beginning of the institutional PA program and one year after graduation (three years after the institutional PA program) to observe the impact of working life on her physical activity behaviour. She was informed about the aim of the study and gave her written consent for the use of the data. All procedures were conducted according to the Declaration of Helsinki.

### 2.1. Institutional PA program for Physically Inactive Students

The program was proposed to every physically inactive student of the university. It consisted of 16 weeks of progressive training that was composed of four blocks of four weeks with 2–3 sessions/week. Each session included a strength part followed by an endurance part. Strength training contained six exercises: half-squat, bench press, leg extension, seated hamstring, rowing, and lunges. Endurance training was performed on a cycloergometer. Table 1 presents the details of the training plan. During the last week of each block, a theoretical course on training methodology was given. Students learned how to plan their training, how to build their strength and endurance sessions, and how to monitor their training load.

### 2.2. Specific Cycling Training Program

After her selection for the RAAM, the participant received a racing bike and followed an 18-week specific cycling training program along with the seven other members of the team. At that time, she had never ridden an outdoor bike. The program was composed of 3–5 sessions/week: 0–1 strength session and 3–5 bike sessions. Six weeks before the RAAM, she participated of her own will in an eight-day pilgrimage, during which she walked a total of 297 km (33.0 ± 8.3 km/day).

### 2.3. Race across America

The team finished the RAAM in 7 days, 12 h, and 46 min (average speed of 25.8 km/h) [13]. The participant completed eight relays for a total of 300 km (37.5 ± 17.1 km/relay) at an average speed of 26.7 km/h. She did not suffer from any conditions during the race, except for some digestive disorders and pains >30 mm on a visual analogue scale in the hip–low back and cervical areas following her two last relays. These disappeared one week after the RAAM.

### 2.4. Monitoring of Workout Load

During the institutional PA program, the specific cycling training program, and the RAAM, the participant was asked to monitor her training load. For endurance parts, the load corresponded to session duration multiplied by the rating of perceived exertion (RPE) [16,17]. For strength parts, it corresponded to the total number of repetitions multiplied by the session RPE [18,19,20]. The sum of both parts gave the total workout load. Figure 2 presents the load of each week (the sum of each workout of the week).

### 2.5. Follow-Up Period

After the RAAM, the participant did not receive any more coaching tips. She was left free to keep on participating in PA or not, and she was not asked to monitor her training load.

### 2.6. Physiological Health Parameters

Physiological health data were collected six times (T1–T6) during the study period (Figure 1) and consisted of anthropometric parameters, a self-reported measure of PA, and strength and endurance parameters.

The anthropometric parameters (the body mass index (BMI) and the waist-to-hip ratio (WHR)) were assessed by using the American College of Sport Medicine guidelines [21]. The self-reported measure of PA was obtained by using the French versions of the international PA questionnaire long form “usual week” [22]. Total PA, subtotals of walking, moderate- and vigorous-intensity PA, and estimated time spent sitting per week were reported. Quadriceps strength (QS) was measured concentrically at 60°/s by using an isokinetic dynamometer. Finally, VO_2_max was assessed during a maximal-intensity, graded exercise test on a cycloergometer.

### 2.7. Psychological Health Parameters

The psychological health data consisted of mood states and self-determination in sports. They were collected six times (T1–T6) during the study period (Figure 1).

To investigate the evolution of mood states during the study period, we used the French version of the profile of mood state questionnaire (POMS), which measures one positive (vigour) and five negative mood states (tension, depression, anger, fatigue, and confusion) [23]. Self-determination in sports was assessed by using the French version of sport motivation scale, which measures three types of intrinsic motivation (IM) (IM to know, IM to accomplish, and IM to experience stimulation), three types of extrinsic motivation (identified regulation, introjected regulation, and external regulation) and amotivation [24].

### 2.8. Qualitative Aspects

We conducted a semi-directed interview at T6. We chose to use conversational material and qualitative approaches in this context so that we could better understand and interpret the participant’s experience from the moment she stepped into the institutional PA program for physically inactive students to the end of the follow-up period. The initial stage allowed for a discussion of the main themes, after which we developed a grid to conduct the semi-directed interview [25].

## 3. Results

### 3.1. Physiological Health Parameters

At T2, the BMI increased from 24.2 to 24.8 kg/m^2^, while the WHR decreased from 0.75 to 0.71. At T3, the BMI and the WHR remained stable at 25.0 kg/m^2^ and 0.72, respectively. At T4, the BMI and the WHR remained stable at 24.9 kg/m^2^ and 0.72, respectively. At T5, the BMI and the WHR increased to 25.3 kg/m^2^ and 0.74, respectively. Finally, at T6, the BMI remained stable at 25.2 kg/m^2^, while the WHR decreased to 0.72. The evolution of the anthropometric parameters is presented in Figure 3.

At T1, a total PA of 83 MET-min/week (composed solely of walking activities) and a sitting time of 3360 min/week were reported. At T2, total PA increased to 4329 MET-min/week (composed of 50%, 48%, and 2% of vigorous-intensity, walking, and moderate-intensity activities, respectively) and sitting time decreased to 2940 min/week. At T3 (excluding the pilgrimage period), total PA increased to 5796 MET-min/week (composed of 50%, 26% and 24% of vigorous-intensity, moderate-intensity and walking activities, respectively), while sitting time decreased to 1560 min/week. At T4, total PA and sitting time peaked at 6720 MET-min/week (composed solely of vigorous-intensity activities) and 4200 min/week, respectively. At T5, total PA decreased to 5043 MET-min/week (composed of 53%, 33% and 14% of walking, moderate-intensity, and vigorous-intensity activities, respectively), while sitting time decreased to 3300 min/week. Finally, at T6, total PA decreased to 4101 MET-min/week (composed of 23%, 46% and 31% of vigorous-intensity, walking, and moderate-intensity activities, respectively), while sitting time decreased to 270 min/week. Figure 4 presents the self-reported measures of PA.

At T2, QS and VO_2_max increased from 1.30 to 1.68 Nm/kg and from 29.2 to 33.9 mL/min/kg, respectively. At T3, VO_2_max increased to 37.8 mL/min/kg, while QS decreased to 1.57 Nm/kg. At T4, VO_2_max decreased to 35.0 mL/min/kg. For logistical reasons, QS was not tested. At T5, VO_2_max increased to 38.1 mL/min/kg, while QS remained stable at 1.56 Nm/kg. Finally, at T6, QS and VO_2_max decreased to 32.0 mL/min/kg and to 1.53 Nm/kg, respectively. Figure 5 presents the evolution of the strength and endurance parameters.

### 3.2. Psychological Health Parameters

During the whole study period, the five negative mood states remained low and stable. Only fatigue increased from 3 to 9 points at T4. The positive mood state remained high during the whole study period. It increased progressively at T2 and T3 from 23 to 26 and to 28 points, respectively. Figure 6 presents the results of the mood states.

At T2, IM to know, IM to accomplish, and IM to experience stimulation increased from 18, 16 and 9 to 25, 26 and 20 points, respectively. Identified and introjected regulation remained stable from 19 and 23 to 19 and 22 points, respectively, while external regulation and amotivation decreased from 9 and 13 to 4 and 5 points, respectively. At T3, intrinsic motivation, extrinsic motivation and amotivation remained stable. At T4, IM to know, IM to accomplish, IM to experience stimulation, and identified regulation peaked at 28 points, while introjected regulation, external regulation and amotivation remained stable at 24, 4 and 4 points, respectively. At T5, IM to know, IM to accomplish, IM to experience stimulation, and identified regulation decreased to 21, 25, 18 and 19 points, respectively. Introjected regulation increased to 27 points, while external regulation and amotivation remained stable at 4 points. Finally, At T6, intrinsic motivation, extrinsic motivation (except for external regulation, which increased to 7 points) and amotivation remained stable. Figure 7 presents the results of self-determination in sports.

### 3.3. Qualitative Aspects

Before entering the institutional PA program for physically inactive students, the participant had bought, on several occasions, a gym membership with the aim to “improve her body image,” but she lacked motivation and time, and she ended up “paying for nothing.”

When the institutional PA program for physically inactive students was offered by the university, she was interested and motivated to have access to a specific and personalized program, as well as by the commitment to record and report on her training, knowing that a dedicated person was available in case of doubts or questions. Her priority at this stage was to be able to organize herself in regard to her studies and personal obligations. One key facilitator for her was to have access to a gym within the university’s walls and to be required to record her training loads during the program. In addition, this appeared to help her motivation to have someone track her training. She declared as having had pleasure to go to training session after some time, which came as a surprise to someone who had never participated in sports before. She did not remember negative feelings or experiences when thinking of the institutional PA program.

The final decision to accept her selection in the university RAAM team was difficult to take, since it was very distant from her initial participation goal. In the end, it was the following words from a team member which convinced her to change her mind, become a team member herself, and partake in the specific cycling training program:
“A “no” must never come from us. If we must have a “no,” it must come from outside of us and not from within. We shouldn’t shoot ourselves in the foot or exclude our own selves.”

She clearly expressed that, throughout the specific cycling training program, she never realized what the RAAM was going to be like. While the other members were projecting themselves and visualizing the race, she was stuck in a form of denial. She went through an “intense” phase when exposed to road cycling and had to familiarize herself with the technical aspects of cycling four months before the RAAM. The specific cycling training program was a lot harder for her compared to the institutional PA program, both physically and mentally. Up until a month before the race, having to follow the training plan and to focus on her studies spared her little or no time to look into what the RAAM actually was. Her character seems to have played an important role for her motivation; she stated that she tends to push forward and “face adversity.” Her true focus during training was the program and not the RAAM itself. Her goal was to sufficiently improve in order to be ready. During that time, she felt the gap between her physical fitness and that of the other team members. She could not participate in all the outdoor training rides with the team, but she managed to keep her motivation high by organizing training sessions with people outside the program.

Her experience at the RAAM was a mix of “pure adrenaline and joy” and “somewhat scary” at the same time. She remained in a denial state as the stages went by, especially regarding altitude elevation, which reassured her and helped her to be more confident. Just like during the institutional PA program, and despite the race difficulty, she had a lot of pleasure during the RAAM stages. She always felt the support of the team beyond the difference in fitness level compared to the rest of them, but she wished she could have had a second person with her fitness background from the initial institutional PA program.

Following the RAAM, she bought a city bike for her daily mobility and regularly did two concurrent trainings per week in the fitness room of the university and then in an outside gym after successfully finishing her bachelor’s in nursing sciences a year after the RAAM. She was determined not to drop off her sports routine, which she had set up with the goal to have fun and stay motivated. The performance logic and pressure that came with it were definitely gone.

Nowadays, her occupation as a nurse with irregular working hours has been identified as an obstacle to her physical activity at the gym, although she keeps riding her bike for transportation for most trips. She remains very conscious of the importance of physical activity and feels the need and want to stay physically active for her own wellbeing. 

## 4. Discussion

The aim of this mixed-method case report was to present the quantitative and qualitative aspects of a sedentary and physically inactive student who participated in the RAAM in a team of eight riders and who was followed-up for the next 28 months. The results showed that the four periods of the study had an important impact on the participant’s life and behaviour. Each period is discussed in the following sections.

### 4.1. Institutional PA Program for Physically Inactive Students

Even if she still spent a lot of time sitting during this period, her level of PA largely increased, reaching, on two occasions, the mean total weekly PA observed in the European Union [26]. This was accompanied by large improvements in her strength and endurance parameters, in accordance with previous results on concurrent strength and endurance training [27]. Before the program, her VO_2_max was lower than previously observed in sedentary nursing students [11]. Following the institutional program, it increased to the same level. Regarding the anthropometric parameters, her BMI slightly increased and her WHR decreased. This result is interesting, since the WHR has been shown to have a higher association with myocardial infarction risk than the BMI [28]. Finally, her intrinsic motivation to participate in sport largely increased during this period, while her external regulation and amotivation decreased, leading to a possible behaviour change [29].

Responses to exercise training are not consistent among all individuals: Some respond well (as was the case in this report), and others respond poorly. Age, sex, and ethnic origin do not appear to be major determinants of responses to exercise training, while genetic background is a strong contributor to inter-individual variation [30,31,32]. One key question is whether the response pattern in a given individual is specific to the given exercise mode and regimen [32]. This interrogation renders the individualization of training protocols complicated. However, it is interesting to note that PA patterns characterized by only one or two sessions/week of moderate or vigorous-intensity PA may be sufficient to reduce risks for all-cause, cardiovascular disease, and cancer mortality, regardless of adherence to PA guidelines [33]. Finally, we must discuss the “risk paradox” of exercise. Despite the fact that long-standing participation in vigorous-intensity PA is associated with a risk reduction of morbidity and mortality, each training session acutely increases the risk of nonfatal cardiovascular events or sudden cardiac death [34]. Even if the proportion of myocardial infarctions that are linked to physical exertion is very low (especially in healthy adults and adolescents), pre-participation exercise testing in individuals considered to be at moderate risk is recommended [21]. It is relevant to note that the latter could already elicit substantial improvements with low-intensity PA [34].

The present program was developed in the university of health sciences to promote increases in PA participation during the academic curricula. From the students’ point of view, it is essential to consider accessibility, flexibility and price in order to facilitate the participation and adherence to this type of program [35]. The proposed program (progressive training and theoretical courses) was free of charge, training sessions were performed in flexible hours in the main building of the university, and the schedule was based on the academic curricula, all of which led to the high participation rate (97%) of the participant. The latter came from the largest group of health providers: the nurses. This program may be relevant to condition their health behaviour and their role as health-promoting actors, since it has been shown that nurses who are more physically active and who receive formal training in this field are better at promoting PA in their clinical practice [12].

### 4.2. Cycling Specific Training Program

Following this period, a stabilisation in her strength and a linear improvement in her endurance parameters (10% higher than sedentary nursing students [11]) were observed, while only low modifications were observed over the other tested parameters. Training was more specific to cycling during this period. However, since the participant had never ridden a bike outdoors, she first had to learn the technical aspects before being able to ride safely. Following this learning phase, she participated in all planned training sessions (but sometimes indoors instead of outdoors), leading to an increased weekly PA. Since we know that motivation can influence athletes’ performances, it would be interesting to measure to what extent her perceived difference in physical fitness compared to the other team members influenced the evolution of her own physiological parameters [36,37].

### 4.3. Race Across America

Even if she only made 6% of the RAAM distance versus >13% for the rest of the team, the participant did it at about the same speed as her teammates without major health issues. She reached her highest intrinsic motivation during the RAAM. This may lead to the hypothesis that, in some individuals, a well-prepared athletic challenge might contribute to the improvement of long-term adherence to PA participation, while this could have the opposite effect in others. An individualised approach should be considered in future interventions that aim to improve PA participation. Finally, she reached her highest level of PA during the RAAM, but she also reached her highest sitting time. This has been observed in individuals who are highly active and participate in sports [38].

### 4.4. Follow-Up Period

Following the RAAM, her main behaviour change was observed: She bought a bike for her daily mobility and practiced regular, concurrent fitness training. The possibility to do sports at her study facility or during her transportation time remained a key driver of her success in maintaining a regular physical activity [35]. In line with these aspects, her psychological health parameters remained stable, whilst her physiological parameters only slightly decreased compared to the end of the institutional PA program (though they remained higher than before the beginning of the study). Her improved self-efficacy and the pride of having completed such a challenge might have contributed to the maintenance of her active behaviour [39].

### 4.5. Limitations

The limitations of this study include that we evaluated only one participant. Thus, our results are not generalizable, but they contribute to a deeper understanding of the topic. In addition, it would have been interesting to explore the qualitative aspects at T1.

## 5. Conclusions

This case report shows that sedentary and physical inactive behaviours can be reversible. Institutional PA programs that include training education in addition to concurrent strength and endurance training can lead to physiological and psychological health benefits. Moreover, in some individuals, an organised athletic challenge can improve motivation and long-term adherence to PA participation. An individualised approach should be considered in future interventions that aim to promote PA participation. Finally, in the specific context of a university of health sciences, we feel that this kind of initiative can positively influence health in the general population by empowering students to become ambassadors for PA promotion.

## Figures and Tables

**Figure 1 ijerph-17-00502-f001:**
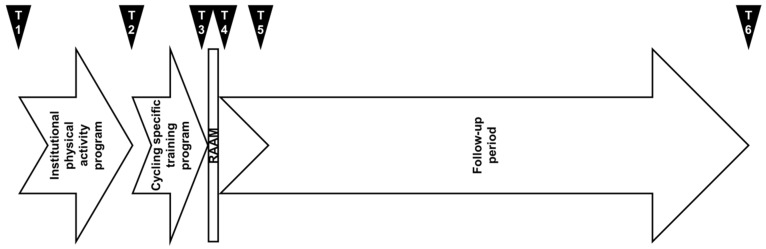
Four periods of the study, which was conducted during a 38-month period. Physiological and psychological health parameters were collected six times over the study period (T1–T6). Qualitative aspects were explored at T6.

**Figure 2 ijerph-17-00502-f002:**
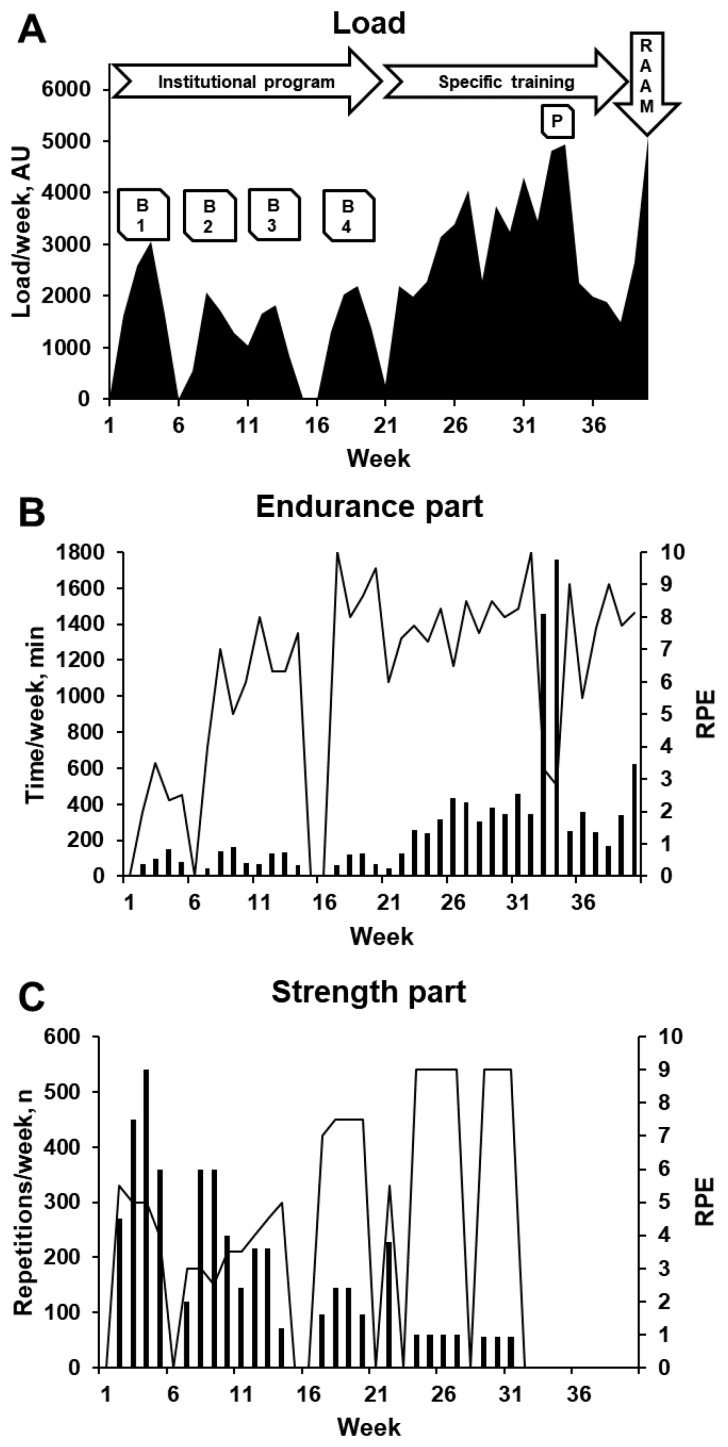
(**A**) Load of each week (the sum of loads of each workout of the week) during the four blocks (B1–B4) of the institutional physical activity program for physically inactive students, the cycling specific training program (with the pilgrimage of eight days (P)), and the Race Across America (RAAM). (**B**) Volume (total time per week) and intensity (mean rating of perceived exertion (RPE) per week) of endurance parts. (**C**) Volume (total number of repetitions per week) and intensity (mean RPE per week) of strength parts.

**Figure 3 ijerph-17-00502-f003:**
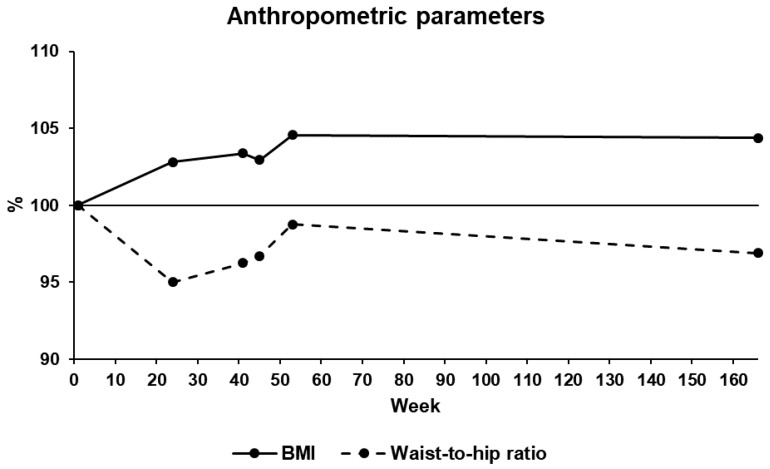
Evolution of anthropometric parameters during the 38-month period of the study. BMI: body mass index.

**Figure 4 ijerph-17-00502-f004:**
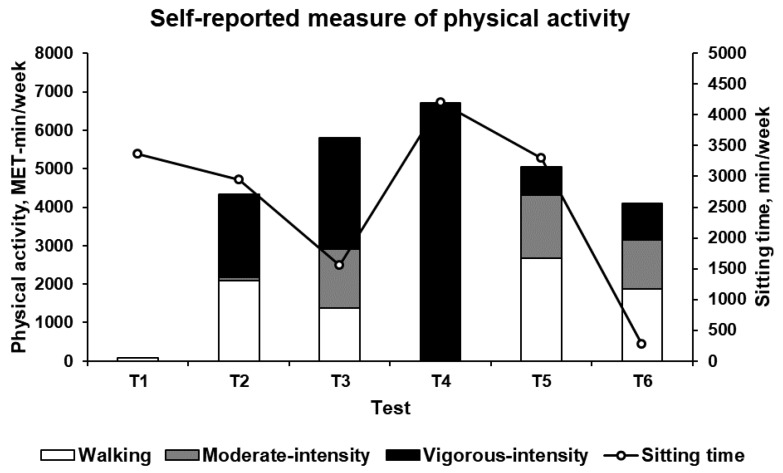
Self-reported measure of physical activity (PA) during a usual week before the six test times (T1–T6). Total PA was composed of walking, moderate-intensity and vigorous-intensity PA.

**Figure 5 ijerph-17-00502-f005:**
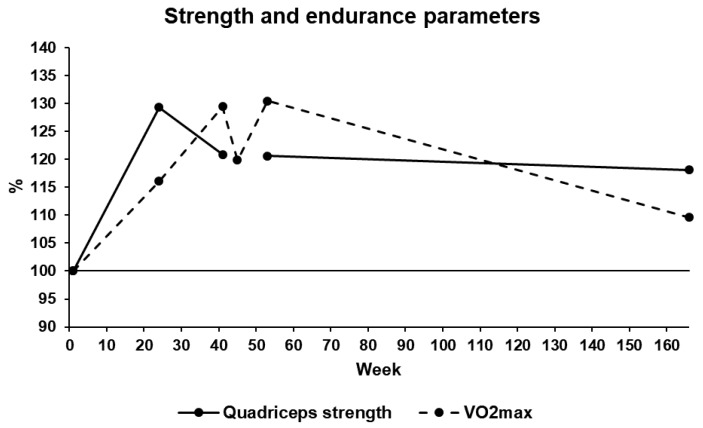
Evolution of strength and endurance parameters (quadriceps strength and maximal oxygen uptake (VO_2_max)) of the participant during the 38-month period of the study.

**Figure 6 ijerph-17-00502-f006:**
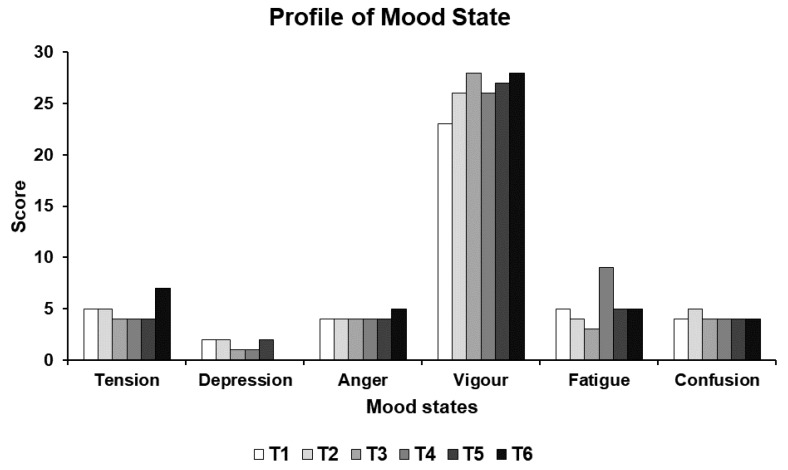
Results of mood states at the six test times (T1–T6).

**Figure 7 ijerph-17-00502-f007:**
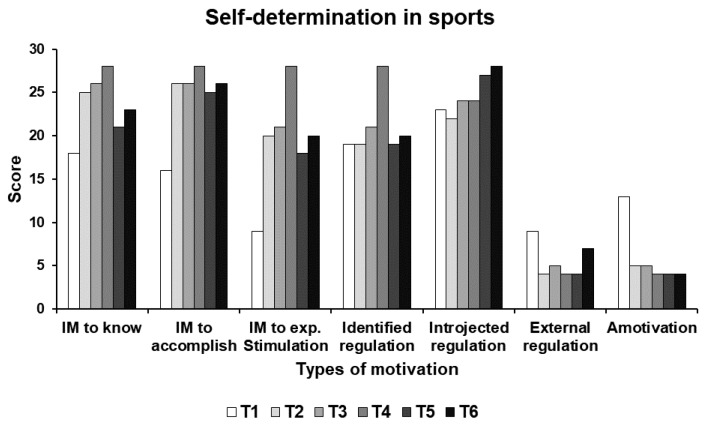
Results of self-determination in sports at the six test times (T1–T6). IM: intrinsic motivation; exp.: experience.

**Table 1 ijerph-17-00502-t001:** Training plan of the institutional physical activity program for physically inactive students.

Block	Parameter	Strength Part	Endurance Part
1	Volume Intensity Rest	1–3 × 15 reps 50%–60% 1-RM 1 min	30–50 min 40%–60% MAP – 70%–85% MHR
2	Volume Intensity Rest	2–3 × 10 reps 70%–80% 1-RM 2 min	2–6 × 4–5 min 75%–80% MAP – 92%–96% MHR 4–5 min
3	Volume Intensity Rest	2–3 × 6 reps 85% 1-RM 3 min	2–3 × 6–9 × 30 s 100% MAP – 96%–100% MHR 30 s/5 min
4	Volume Intensity Rest	2–3 × 4 reps 90% 1-RM 3 min	3–4 × 4–6 × 6 s >250% MAP – 90%–95% MHR 6 s/5 min

1-RM: one repetition maximum; MAP: maximal aerobic power; MHR: maximum heart min, minute(s); rate; reps, repetitions; sec, second(s).

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
