# Peer review of "From Sedentary and Physical Inactive Behaviours to an Ultra Cycling Race: A Mixed-Method Case Report"

_ijerph, 2020, doi:10.3390/ijerph17020502_

Round 1
Reviewer 1 Report
The manuscript entitled “From sedentary and physical inactive behaviours to the ultra cycling Race Across America (RAAM): a mimed-methods case reports” deals with the assessment on a nursing student who participated in a high-demand cycling contest. At the beginning, the student was considered herself as a “not physical activity” person. After the competition, however, the student and the studies that were carried out onto her concluded that she was ready to increase the physical activity and even she bought a mountain bike to continue the physical exercises.
Some issues have to be addressed before the manuscript can be considered as suitable for publication:
In my opinion, the contest RAAM has to be removed from the title, since is just a part of the study, but is not essential to understand the method applied in the article. A further discussion has to be carried out. The study is based in only one person. In medical sciences, especially, an analysis based in one person is not strong enough to achieve reliable conclusions. The structure of the abstract has to be modified. The use of captions such as background and conclusions are not necessary in this section of the manuscript. I guess there is a typo in the title. I don’t know if it should be “mixed-method”. The same for the manuscript, use mixed-method with a dash.
Author Response
Reviewer #1
The manuscript entitled “From sedentary and physical inactive behaviours to the ultra cycling Race Across America (RAAM): a mimed-methods case reports” deals with the assessment on a nursing student who participated in a high-demand cycling contest. At the beginning, the student was considered herself as a “not physical activity” person. After the competition, however, the student and the studies that were carried out onto her concluded that she was ready to increase the physical activity and even she bought a mountain bike to continue the physical exercises.
Some issues have to be addressed before the manuscript can be considered as suitable for publication:
In my opinion, the contest RAAM has to be removed from the title, since is just a part of the study, but is not essential to understand the method applied in the article.
The title was changed as follows (lines 2-4):
From Sedentary and Physical Inactive Behaviours to An Ultra Cycling Race: A Mixed-Methods Case Report
A further discussion has to be carried out. The study is based in only one person. In medical sciences, especially, an analysis based in one person is not strong enough to achieve reliable conclusions.
We thank the reviewer for his/her comment. We have added this aspect in the limitations part (lines 338-339):
Limitations of this study include that we evaluated only one participant. Thus, our results are not generalizable, but they contribute to a deeper understanding of the topic.
The structure of the abstract has to be modified. The use of captions such as background and conclusions are not necessary in this section of the manuscript.
As suggested, we have removed captions in the abstract (lines 24-38):
In faculties of health sciences, almost 30% of nursing students exercise less than once a week. This mixed methods case report presents the 38-month evolution of physiological and psychological health parameters of a sedentary and physically inactive nursing student. During this period, she first took part in a one-semester institutional physical activity (PA) program offered by her University, before being selected for participation in the Race Across America (RAAM) with a University relay team. In the four months leading up to RAAM, she followed a cycling training program. After the RAAM, she was followed-up for the next 28 months. Results show that each phase of the study had an important impact on the subject and showed that sedentary and physical inactive behaviours are reversible. Institutional PA programs, including training education in addition to concurrent strength and endurance training can lead to physiological and psychological health improvements. For some individuals, participating in an athletic challenge can improve motivation and long-term adherence to PA participation. An individualised approach should be considered in future interventions aiming to promote PA participation. In the specific context of a University of Health Sciences, this kind of initiative could positively influence the general population’s health, by empowering students to become role models towards PA promotion.
I guess there is a typo in the title. I don’t know if it should be “mixed-method”. The same for the manuscript, use mixed-method with a dash.
The title was changed as follows (lines 2-4):
From Sedentary and Physical Inactive Behaviours to An Ultra Cycling Race: A Mixed-Method Case Report
And in the manuscript (lines 25-26, 67-69, 267-269):
This mixed-method case report presents the 38-month evolution of physiological and psychological health parameters of a sedentary and physically inactive nursing student.
The aim of the present mixed-method case report was to present physiological and psychological health parameters of a sedentary and physically inactive student who participated in the RAAM with the University team.
The aim of this mixed-method case report was to present quantitative and qualitative aspects of a sedentary and physically inactive student, who participated in the RAAM in a team of eight riders and who was followed-up for the next 28 months.

Reviewer 2 Report
The main problem is that only a subject has been studied, as case report publication the paper is too long, my opinion is that a simple case could be presented in another way.
The study presents an absolutely predictable evolution, on a single case, an initial study on the personality traits of the person studied is missing
The sequence of scans carried out needs a justification in the time intervals
It would be important to know the degree of motivation of the person about what the intervention is done. From a practical point of view, the protocol used should be based and justified, not only from a global perspective, but if there are objective reasons that have forced to customize the intervention in this specific case
Author Response
Reviewer #2
The main problem is that only a subject has been studied, as case report publication the paper is too long, my opinion is that a simple case could be presented in another way.
The study presents an absolutely predictable evolution, on a single case, an initial study on the personality traits of the person studied is missing
We thank the reviewer for his/her comment. Indeed, an initial study on the personality traits of the person being studied is missing. It was not foreseeable that this student would be selected to participate in the RAAM before the study began. For this reason, we did not conduct qualitative measures prior to the study with her. However, the study provides data about her mood and motivation prior to the start of the institutional program. Moreover, for us, it was not predictable that a girl who had never ridden a bike before would end up buying a bike to continue the physical exercises.
We have added this aspect in the limitations part (line 339):
In addition, it would have been interesting to explore the qualitative aspects at T1.
The sequence of scans carried out needs a justification in the time intervals
As requested, we have added the following sentence (Lines 83-87):
Follow-up tests were performed two (T5) and 28 months (T6) after the RAAM, in order to monitor the participant's progression one year after the beginning of the institutional PA program and one year after graduation (three years after institutional PA program) to observe the impact of working life on her physical activity behaviour.
It would be important to know the degree of motivation of the person about what the intervention is done.
As explained in qualitative aspects section, the participant was interested and motivated to have access to a specific and personalized program, and by the commitment to record and report on her training, knowing that a dedicated person was available in case of doubts or questions. She was not particularly motivated by the study aspects but she accepted to participate to each tests phase.
Behaviour change for physical activity is a highly individual and multimodal intervention. There is no global consensus for theses protocols, but we can admit nowadays that the principles of behavioural economics imply the following aspects: facilitation, education, experiencing, socializing, and competence/confidence development. These aspects were part of the protocol. As the reviewer points out, it is important to consider the specificity of the case and to adjust accordingly, whilst applying global principles of training of motivational change.

Round 2
Reviewer 1 Report
The authors have correctly addressed my comments.
Reviewer 2 Report
The paper is suitable for its publication in the last version